



# Development of a cascade impactor optimised for size-fractionated analysis of aerosol metal content by total reflection X-ray fluorescence spectroscopy (TXRF)

Claudio Crazzolara[1,2], Andreas Held[1]

[1]Environmental Chemistry and Air Research, Technische Universität Berlin, Berlin, 10623, Deutschland
[2]Bruker Nano GmbH, Berlin, 12489, Deutschland

*Correspondence to*: Claudio Crazzolara (crazzolara@tu-berlin.de)

**Abstract.**

A new cascade impactor has been developed with the arrangement of the classifying nozzles optimised for analysis of the collected particles by total reflection X-ray fluorescence (TXRF). TXRF offers detection limits in the range of a few pg of absolute mass and therefore poses great potential for the elemental analysis of heavy metals in aerosol particles. To fully exploit this high sensitivity, particles have to be collected in the effective analysis area of the TXRF instrument, which is often smaller than typical deposition patterns of commercial impactors or filter samplers. This is achieved by a novel, compact arrangement of the classifying nozzles within a circular area of a diameter of less than 5 mm. A decreasing density of the nozzle spacing from the inside to the outside of the nozzle cluster allows for constant cross flow conditions, minimising the mutual influence of the individual nozzles. The design of a multi-stage cascade impactor is presented, to individually sample PM10, PM2.5 and PM1 size fractions. Considering the high sensitivity of TXRF analysis, constructive measures have been taken to prevent attrition of impactor material which might lead to methodical blank values. Experimental validation confirms that neither attrition nor cross-contamination can be observed. Furthermore, a new spin-coating method has been developed which makes it possible to apply a thin and defined adhesive layer of grease to the sample carrier with good repeatability. Application of the impactor in a case study at an urban site at Potsdamer Platz, Berlin, Germany shows that at a moderate sampling volume flow rate of 5 litres per minute, the particle mass collected in 30 minutes or less is sufficient for reliable TXRF analysis of heavy metal concentrations (Fe, Zn, Cu, Mn, Pb, Ni) in ambient aerosol. This high time resolution enables snapshot sampling, e.g. to quantify variations in particle source strengths. Overall, the new impactor optimised for TXRF analysis bears great potential to improve the quantification of particulate trace metals and other elements in PM10, PM2.5 and PM1 with high time resolution.

## 1 Introduction

Air pollution caused by aerosol particles has detrimental impacts on humans, animals and plants (Fuzzi et al. 2015). Airborne particles with an aerodynamic diameter of less than 10 µm have been associated with adverse health effects in numerous studies (Chen & Hoek 2020). Fine particles with an aerodynamic diameter of less than 2.5 µm can penetrate deep into the body via the respiratory tract, and are generally associated with a greater risk potential than coarse particles (e.g. Feng et al. 2016). Therefore, the World Health Organization has recommended, and many countries have imposed legal limits for the mass concentration of health-relevant size fractions of particulate matter such as PM10 and PM2.5 (e.g. WHO 2021, European Parliament 2008). In addition, the chemical composition of aerosol particles is important with regard to element-specific hazard potentials (Corriveau et al. 2011), and target values for the mass content of certain elements have been introduced (European Parliament 2004). The European standard method EN 14902:2005 for measuring lead, arsenic, cadmium and nickel in the PM10 fraction requires particle collection on filters, sample digestion and elemental analysis by graphite furnace atomic





absorption spectroscopy or inductively coupled plasma mass spectrometry (CEN, 2005). This offline approach is rather time-consuming and does not allow for studying transient concentration changes with high time resolution.

Fast and sensitive elemental analysis of size-fractionated aerosol samples is possible with a combination of impactor sampling and total reflection X-ray fluorescence (TXRF) spectroscopy. Impactor sampling of aerosol particles is commercially available and widely used (Marple 2004). Elemental analysis of collected aerosol particles has often been done by inductively coupled plasma mass spectrometry (e.g. Gietl et al. 2010) or X-ray fluorescence (XRF) (e.g. Kuhn et al. 2005) and TXRF (e.g. Schneider 1989). In XRF and TXRF, the sample is excited with X-ray radiation, and the resulting fluorescence is characteristic

of individual chemical elements in the sample. In TXRF, the sample is prepared on a flat sample carrier with a polished surface. A polychromatic X-ray beam is monochromatized and irradiates the sample at a very shallow angle of approximately 0.1°, which leads to total reflection of the incident X-ray beam at the surface of the sample carrier. Fluorescence radiation from the sample is analysed with a detector located directly above the sample with a large solid angle of view. The effective analysis area results from the superposition of the area that is excited by the X-ray beam and the field of view of the detector. Since

there is almost no interaction between the exciting X-ray beam and the substrate of the sample carrier, the signal-to-noise ratio is significantly improved compared to conventional XRF. As a result, the detection limit of TXRF is superior to XRF (Yoneda and Horiuchi, 1971), and can reach down to a few picograms of absolute mass on the sample carrier substrate (Streli 2006). Recently, Prost et al. (2017) and Seeger et al. (2021) demonstrated the huge potential of TXRF analysis for the elemental analysis of aerosol particles collected with a commercial impactor, in particular for sampling times of a few hours only. Despite

promising results, commercial impactors are not fully optimised for TXRF analysis: The area on the sample carrier in which the classifying nozzles deposit the particles is usually significantly larger than the area analysed by TXRF. As a result, only a fraction of the particles collected by the impactor are analysed, and consequently, the overall sensitivity is reduced. To take full advantage of the high detection sensitivity of TXRF analysis, it is necessary to analyse the entire sample collected by the impactor.

Here, we present the development of a cascade impactor optimised for size-fractionated analysis of aerosol metal content by TXRF. The target application of the newly developed impactor is to quantify low concentrations of heavy metals in atmospheric aerosol samples collected over periods of one hour or less. The impactor is designed (1) to collect particles in individual size fractions to quantify the metal content in PM10, PM2.5 and PM1 separately, (2) to collect particles in circular areas with a diameter of approximately 5 mm or less in the centre of the sample carrier for full TXRF analysis, and (3) to

provide low blank values and minimum cross-contamination between subsequent sampling periods. We will first present the general impactor design, the nozzle arrangement and an improved coating method for sample carriers. After describing the experimental procedures to validate the impactor performance and the first application of the impactor in outdoor air, we will present and discuss the experimental results.

## 2. Impactor development

### 2.1 General impactor design

For size-fractionated elemental analysis of PM10, PM2.5 and PM1, the relevant particle size fractions must be collected individually. At the same time, particles should be collected on as few impactor stages as possible, to simplify handling and to increase the absolute particle mass on individual stages. These considerations imply a configuration of three stages with 50 % separation diameters of 10.0 µm, 2.5 µm and 1 µm, respectively, and a fourth stage with a separation diameter well below

1µm, so that almost all particles contributing to PM1 in terms of mass are collected. Previous TXRF analysis results of impactor samples (e.g. Seeger et al., 2021) suggest that particles of 0.1 µm diameter or less make only a minor contribution to particulate mass. Therefore, a fourth stage with a separation diameter of approximately 0.1 µm can be used to collect almost the entire



PM1 fraction. In the present study, we apply a fourth stage with a nominal separation diameter of 0.13 µm, and a fifth stage
with a nominal separation diameter of 0.095 µm for experimental purposes.

An optimised impactor design allows particle collection directly on the sample carriers for TXRF analysis, providing compact
deposition patterns in circular areas with diameters less than 5 mm. This compact deposition pattern is achieved by choosing
a low flow rate of 5 standard litres per minute (slm, equivalent to volume flow at standard conditions, 1013 hPa, 20 °C), which
allows single classifying nozzles to be used on stage 1 and 2 (10 µm and 2,5 µm separation diameter), and multiple nozzles
located in a circular area with a diameter of less than 5 mm on stages 3, 4 and 5, without exceeding a critical Reynolds number
of 3000.

Each impactor stage has an upper body accommodating the classifying nozzles and a lower body accommodating the sample
carrier (Figure 1a). The classifying nozzles are incorporated in an exchangeable nozzle module (Figure 1b), which defines the
number, diameter $d_n$, length $l_n$ and lateral separation $d_{lat}$ of the classifying nozzles, as well as the distance $s$ between the end of
the nozzle and the surface of the sample carrier. These parameters have an influence on the separation diameter and the
separation characteristics of the impactor stage. The sample carrier is fixed by means of an elastic mounting ring made from
laser sintered polyamide. Due to this modular design, the number of impactor stages can be easily adapted to different
applications to allow for different separation diameters and size-fractionated samples. When designing the modules, care was
taken to guide the flow through the impactor with as little interference as possible to minimise particle losses.

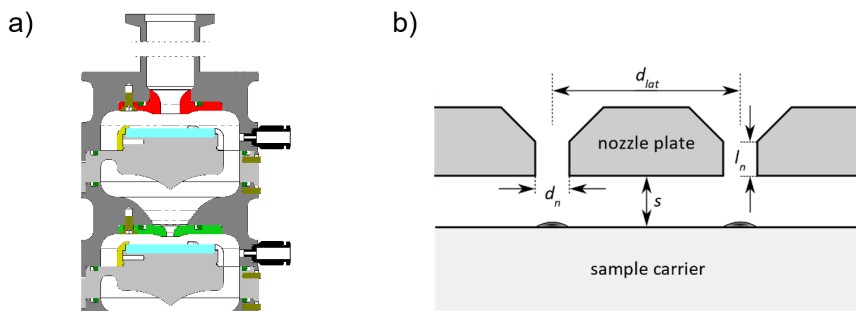

**Figure 1: a) Schematic of the first two impactor stages, with an upper body (dark grey) accommodating the nozzle module (red
and green) and a lower body (light grey) accommodating the sample carrier (light blue).**
**b) Cross-section through a nozzle module with two adjacent nozzles and the sample carrier**

To assemble the impactor, the modules are stacked on top of each other into a support frame to form a cascade. Two push
rod clamps compress the sealing rings between the modules. Care was taken to minimize metal-to-metal friction during
assembly and disassembly of the impactor, which might contaminate samples with metal particles in regard to the high
sensitivity of the TXRF analysis. A locking pin prevents the modules from rotation in stacked state, and a sealing ring
positioned internally, prevents particles from entering the impactor chamber, in case a small amount of attrition would occur
at the metallic mating surfaces during assembly. In addition, the inner surfaces of the impactor have been polished
electrically after machining to minimise surface roughness and facilitate cleaning.

**2.2 Geometry and arrangement of the classifying nozzles**

The geometry and number of the classifying nozzles were designed according to Marple and Willeke (1967). The aerodynamic
50 % separation diameter $d_{ae50}$ of an impactor stage can be expressed as a function of the nozzle diameter $d_n$ (Eq. 1),





$$d_{ae50} = \sqrt{St_{50}} \cdot \sqrt{\frac{9 \cdot \eta \cdot d_n}{\rho_p \cdot v_0 \cdot C_c}} \qquad \text{[Eq. 1]},$$

where $\eta$ is the dynamic viscosity of the air ($\eta$ = 1.81 x $10^{-2}$ g $m^{-1}$ $s^{-1}$ at 20 °C), $C_c$ is the Cunningham slip-correction factor

(Allen & Raabe 1982, 1985), $\rho_p$ is the particle density ($\rho_p$ = 1 x $10^6$ g $m^{-3}$ by definition of aerodynamic diameter), $v_0$ is the

average flow velocity in the nozzle, and $St_{50}$ is the critical Stokes number. For impactor stages with a single circular nozzle, a

value $St_{50}$ = 0.24 can be assumed (Rader & Marple 1985), and for impactors with multiple circular nozzles, a value $St_{50}$ = 0.216

was determined experimentally (Hillamo & Kauppinen 1991). The average flow velocity in the nozzle $v_0$ is a function of the

volumetric flow rate $Q$, the nozzle diameter $d_n$, and the number of nozzles, $N_n$ (Eq. 2):

$$v_0 = \frac{4 \cdot Q}{\pi \cdot d_n^2 \cdot N_n} \qquad \text{[Eq. 2]},$$

It is important to note that while the mass flow rate through the impactor is constant, e.g. 5 slm corresponding to a volumetric

flow rate of 5 litres per minute at standard conditions (1013 hPa, 20 °C), the actual volumetric flow rate $Q$ increases

downstream with each impactor stage as a result of the pressure drop and the associated expansion of the gas. The number of

nozzles $N_n$ per classification stage was selected so that the Reynolds number $Re$, calculated according to Eq. 3, does not exceed

a value of 3000 in each individual nozzle:

$$Re = \frac{\rho_{air} \cdot v_0 \cdot d_n}{\eta} \qquad \text{[Eq. 3]},$$

where $\rho_{air}$ is the density of air ($\rho_{air}$ = 1204 g $m^{-3}$ at 20 °C). In Table 1, the nominal nozzle diameters $d_n$ and the number of

nozzles $N_n$ for each separation stage are indicated along with the calculated separation diameter $d_{ae50}$ acc. Eq.1, the average

flow velocity $v_0$ acc. Eq. 2, the Reynolds number $Re$ acc. Eq. 3, the cylindrical throat length $l_n$, and the distance between the

end of the nozzle and the surface of the sample carrier $s$. At the impactor stages 1 and 2, the classifying nozzles exhibit a

bellmouth-shaped inlet that tapers to the nozzle diameter. The classifying nozzles in stages 3 to 5 have a conical taper at the

inlet with an opening angle of 90 degrees (Figure 1b). Long classifying nozzles favour the development of a parabolic, pipe-

like flow profile, resulting in a less steep separation curve. Therefore, the cylindrical throat length of the classifying nozzles $l_n$

was kept short in order to facilitate the formation of a plug-shaped flow profile. The ratio of the length $l_n$ and the diameter $d_n$

of the cylindrical section of the nozzle is kept between 0.5 (stage 1) and 1.5 (stages 4 and 5). The distance $s$ between the nozzle

end and the sample carrier should be kept as small as possible with regard to the steepness of the separation efficiency curve,

however, it must be sufficiently large to allow the air flow to escape unhindered radially to the direction of flow through the

classifying nozzle (Marple et al. 1991). Short distances also enhance the blow-off of impacted particles. The distances $s$ are

between 4.9 mm (stage 1) and 0.6 mm (stages 4 and 5).


**Table 1: Design parameters of the five separation stages of the newly developed cascade impactor**

| Stage | Nominal nozzle diameter $d_n$ in mm | Number of nozzles $N_n$ | Separation diameter $d_{ae50}$ in µm (acc. Eq. 1) | Average flow velocity $v_0$ in m/s (acc. Eq. 2) | Reynolds number $Re$ (acc. Eq. 3) | Cylindrical throat length $l_n$ in mm | Distance to sample carrier surface $s$ in mm |
|---|---|---|---|---|---|---|---|
| 1 | 6.50 | 1 | 9.96 | 2.5 | 1090 | 3.26 | 4.9 |
| 2 | 2.60 | 1 | 2.47 | 15.7 | 2720 | 1.30 | 2.6 |
| 3 | 0.75 | 7 | 0.915 | 27.2 | 1360 | 1.125 | 0.75 |
| 4 | 0.20 | 19 | 0.13 | 167.7 | 2230 | 0.3 | 0.6 |
| 5 | 0.20 | 24 | 0.095 | 197.3 | 2625 | 0.3 | 0.6 |

In stages 3 to 5, multiple classifying nozzles were required to realize small separation diameters with $Re < 3000$. If the lateral distance $d_{lat}$ between several classifying nozzles is too small, the separation characteristics may be affected. This can occur due to an interference of the outflow of two adjacent classifying nozzles colliding and causing a secondary impaction, resulting
in a "tailing" of the separation curve towards smaller particle diameters (García-Ruiz et al. 2019a,b). Hence, it is important to establish an optimum distance between nozzles that allows a compact deposition pattern, but at the same time does not strongly influence the separation characteristics of the impactor stages. Fang et al. (1991) developed a method for estimating the extent of cross-flow, by calculating the cross-flow parameter $k_{cf}$ according to Eq. 4,

$$k_{cf} = \frac{N_n \cdot d_n}{4 \cdot d_c} \qquad [\text{Eq. 4}]$$

where $N_n$ is the number of nozzles in the nozzle cluster, $d_n$ is the individual nozzle diameter and $d_c$ is the diameter of the nozzle cluster, i.e. the circular area where the nozzles are arranged. Empirical particle collection data show that impactor stages with multiple nozzles operate satisfactorily if the cross-flow parameter $k_{cf}$ is less than a critical value of 1.2 (Fang et al., 1991).

To meet this challenge, we have developed a special arrangement of the classifying nozzles in which the distance between adjacent nozzles increases from the centre of the nozzle cluster towards its outer edge. This ensures that all nozzles are operated
at the same cross-flow conditions and that the radial outflow does not exceed a maximum crossflow velocity. In the classifying nozzles arrangement of stages 3 to 5 ($d_c$ = 3, 3.6 and 4.2 mm) of our impactor, the corresponding cross-flow parameters $k_{cf}$ acc. Eq. 4 are 0.44, 0.27, and 0.31, respectively, thus well below the critical value of 1.2 determined by Fang et al., (1991). An exemplary illustration of the distribution of the crossflow velocity of the nozzle cluster of stage 4 can be found in Figure S1 (Supplementary Material).

Figure 2 shows scanning electron microscopic (SEM) images with perspective views of all 19 classifying nozzles on both sides of the nozzle module of stage 4, and an enlarged view of a single classifying nozzle. On the high-pressure side (Figure 2a,b), the conical sections of the classifying nozzles can be seen, where the diameter tapers from 500 µm to 200 µm in the direction of the gas flow, followed by the circular cylindrical section to the end of the classifying nozzles on the low-pressure side (Figure 2c,d).






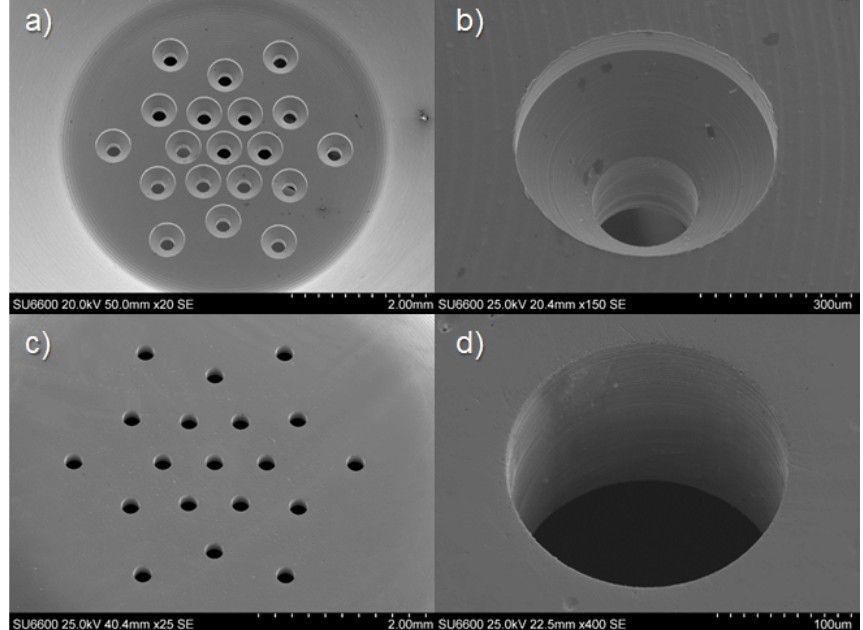

**Figure 2:** Scanning electron microscopic (SEM) images of the nozzle module of stage 4 with perspective views of a) all classifying nozzles and b) a single classifying nozzle on the high-pressure side, and c) all classifying nozzles and d) a single classifying nozzles on the low pressure side.

Measuring the inner diameters of the classifying nozzles with a light-optical measuring microscope (Keyence IM-7000)

resulted in relative deviations from the nominal diameters in the low single-digit percentage range (Table S1 in Supplementary Material). These measurements reflect highly uniform diameters of the individual classifying nozzles of the multi-nozzle impactor stages 3 to 5.

**2.3 Spin-coating of sample carriers**

Depending on the application, it is common practice to apply an adhesive layer to the impaction substrate (Injuk & Grieken

1995). This adhesive layer prevents effects such as bounce-off or blow-off of impacted particles from the impaction substrate. Silicon-based polymers like polysiloxane silicone oils or hydrocarbon-based greases can be used as adhesives. Here, the silicone- and halogen-free ultra-high vacuum grease Apiezon-L (M&I Materials Limited) is utilized, which is characterised by a low vapour pressure, low creep or carry over and high purity. For TXRF analysis, Apiezon-L is virtually blank value-free except for low blank values of sulphur, iron and bromine.

Up to now the adhesive layer has been applied onto the impaction substrate by dabbing or brushing (Injuk & Van Grieken 1995, Streli 2006), by spraying (Seeger 2021) or by impacting out of an aerosol (Hillamo & Kauppinen 1991). For TXRF analysis, it is necessary that the adhesive layer is applied to the sample carriers in a reproducible manner and without local inhomogeneities of the layer thickness. In addition, the layer should be as thin as possible so that it does not induce a background signal in the fluorescence spectrum. Therefore, a spin-coating process was developed, which ensures a uniform

and flat adhesive layer on the sample carrier surface. 10 g Apiezon-L grease are first dissolved in 100 ml toluene. The resulting opaque yellow solution is filtered through a syringe filter with a pore size of 0.2 μm to obtain a clear yellow solution. Then, 15 μl of the filtered, clear-yellow solution are taken up by a pipette and applied to the centre of the sample carrier which is rotating at a speed of 6000 revolutions per minute. The excess solution is spun off and a thin layer of the solution remains on





the sample carrier. Within a short time of approximately 0.5 s, the toluene volatilises, and a homogeneous layer of Apiezon-L
remains on the sample carrier surface. The layer thickness can be varied by the mass fraction of grease in the solution, and by
the speed of rotation of the sample carrier during spin-coating.

## 3 Methods

### 3.1 Experimental procedures to validate impactor performance

#### 3.1.1 Blank values due to particle attrition or adhesive coating

In the process of assembly and disassembly of the impactor, particle attrition can occur due to metal-to-metal friction. If these
particles settle on the collection substrate, a blank value may result during TXRF analysis. Klockenkämper et al. (1995) report
blank values of 40 % on average for stainless steel impactors in relation to the actual measured value. All metal parts of the
impactor developed in the present study are made from chrome-nickel steel (type 1.4301 X5CrNi18-10), which contains mainly
iron and additionally 18 % chromium, 10 % nickel, a maximum of 2.0 % manganese and a maximum of 1 % silicon. The
sealing rings are made of fluoropolymer, the elastic locking rings are made of polyamide and the screws of the nozzle modules
are made of stainless steel. Any attrition could release particles of the above-mentioned composition. To determine the
potential of contamination due to attrition, the following experimental procedure was carried out in a clean-room environment:

First, the impactor was disassembled into its individual parts, cleaned, and dried. After cleaning, sample carriers made from
$SiO_2$ glass were installed. Prior to installation, these sample carriers were cleaned and then subjected to a TXRF analysis
(TXRF no. 1). Regarding this experiment, all TXRF analyses were performed for a duration of 1000 s using Mo-K alpha
excitation. After installation of the sample carriers, the impactor stages were assembled and tensioned with the push rod clamps.
In this state, the impactor would be ready for particle collection. Instead, the push rod clamps were relaxed again, and the
impactor was disassembled, preparing the extraction of the sample carriers for analysis. This assembly-disassembly procedure
was repeated for five times to provoke particle attrition and deposition onto the sample carriers. Afterwards, the sample carriers
were removed and analysed (TXRF no. 2). Subsequently, the sample carriers were spin-coated with an adhesive film and
analysed by TXRF (TXRF no. 3) thereafter, to determine the influence of impurities and possible blank values in the adhesive
coating. The coating is intended to retain any particles that may have formed. Next, the coated sample carriers were mounted,
and the assembly-disassembly procedure was repeated for five times. Continuing, the impactor was operated for five minutes
at a volumetric flow rate of 5 slm, with the air sampled taken from a particle-free environment. This was done to agitate any
particles that may have formed inside the impactor, depositing them on the sample carriers by impaction. The impactor was
then disassembled, the sample carriers removed and analysed using TXRF (TXRF no. 4).

#### 3.1.2 Cross-contamination between subsequent sampling periods

Particles may deposit on the inner surfaces of the impactor during aerosol sampling. In a subsequent sampling operation, these
particles may be resuspended from the wall and deposit on the sample carriers, thus affecting the result of the analysis. The
error caused by this potential "memory" effect may amount to an average of 30 % in relation to the actual value, as reported
by Klockenkämper et al. (1995). Thorough cleaning after each particle collection operation can help to avoid this cross-
contamination error but is time-consuming and a large effort.

To prevent cross-contamination, the flow chamber of our impactor was designed with large curvature radii to avoid abrupt
changes along the flow path as far as possible. In addition, the surface roughness was reduced by electrolytic polishing of
internal surfaces. To investigate the effectiveness of these design measures, the following experiments were carried out:


First, sample carriers (SiO₂ glass) were cleaned and coated with an adhesive film using the spin-coating method described in section 2.3. Subsequently, blank spectra of the coated sample carriers were recorded (TXRF no. 5). For this purpose, the element iron was evaluated, as in many cases it is the most abundant heavy metal in the fine particle fraction. The coated sample carriers were then placed in a cleaned impactor and the impactor was operated for five minutes with filtered, particle-

free air at a flow rate of 5 slm. Next, the sample carriers were analysed again (TXRF no. 6), evaluating whether particles detach from the wall of the flow chamber during the five-minute operation of the impactor. The same sample carriers were installed in the impactor again, and it was operated for 30 minutes with particle-laden atmospheric air and a flow rate of 5 slm. During this collection operation, the particle mass concentration of the atmospheric air was measured in three size fractions (PM10, PM2.5 and PM1) using an optical aerosol spectrometer (Fidas Frog, PALAS GmbH, Karlsruhe, Germany), which had

previously warmed up for 120 minutes and was adjusted using calibration dust (MonoDust, PALAS GmbH, Karlsruhe, Germany). Following the 30-minute collection run, the particle-laden sample carriers were removed from the impactor and subjected to further TXRF analysis (TXRF no. 7). To determine whether subsequent sampling with this possibly contaminated impactor would result in cross-contamination due to resuspension of particles deposited on the walls of the impactor, spectra were recorded from additional, clean and coated SiO₂ glass sample carriers (TXRF no. 8). These sample carriers were then

placed in the potentially particle contaminated impactor, and operated for 15 minutes with filtered, particle-free air at a flow rate of 5 slm. Ultimately the sample carriers were removed from the impactor and subjected to TXRF analysis (TXRF no. 9) to determine whether the particle load of the sample carriers had increased, i.e. whether cross-contamination had occurred.

### 3.2 Particle collection in outdoor air

Particles from atmospheric air were collected on SiO₂ glass sample carriers during three consecutive sampling periods using

the newly developed impactor and a mobile battery-operated pump unit. Due to the low volume flow of 5 litres per minute at standard conditions, the impactor can be operated with a simple diaphragm vacuum pump (N 813.3, KNF Neuberger GmbH, Freiburg, Germany) powered from a rechargeable Li-ion battery. A mass flow sensor (SFM4300-20-P, Sensirion AG, Stäfa, Switzerland) is applied to measure the gas mass flow through the impactor. The sensor is factory-calibrated, has an operating temperature range of -20 to +80 °C and provides a temperature-compensated output signal. Figure 3 shows the impactor and

the pump unit during operation in the field.

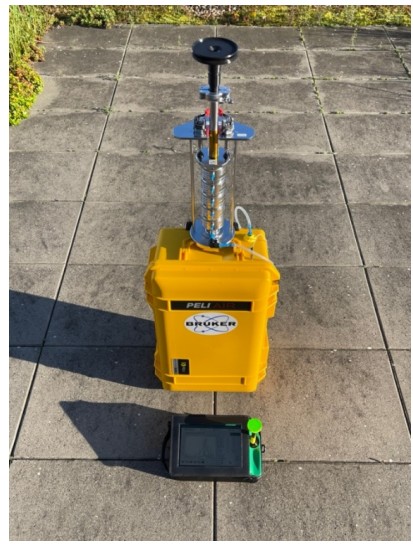





**Figure 3: Impactor with five impactor stages in assembled condition with omnidirectional air intake (black) placed on the yellow transport case containing the pump and a rechargeable battery; Fidas Frog aerosol spectrometer on the ground next to the particle sampling unit.**

The impactor was located next to the road intersection of Potsdamer Platz (52.50926 N, 13.37699 E), an urban roadside environment in Berlin, Germany. Particle collection was carried out on 29 August 2022 in the morning between 8:00 and 9:36 CEST (Central European Summer Time) in three consecutive collection periods, lasting for 30 minutes each. The first period lasted from 8:00 to 8:30, the second period from 8:32 to 9:02, and the third period from 9:06 to 9:36. With an air mass flow rate of 5 slm, a total volume of 150 litres (at standard conditions 293 K, 1013 hPa) was sampled in each individual 30-minute collection period. Between the collection periods, the impactor was reloaded with plain, greased sample carriers, and the loaded sample carriers were packed for transport to the laboratory and subsequent TXRF analysis.

### 3.3 TXRF analysis

For chemical analysis of particles collected on the sample carriers, a Bruker S4 T-STAR TXRF spectrometer (Bruker Nano GmbH, Berlin, Germany) was applied. The main components of the instrument are shown schematically in Figure 4. The TXRF spectrometer comprises two air-cooled X-ray tubes, each with an electrical power of 50 W. In one tube, molybdenum is used as anode material, and tungsten in the other tube. By applying three multilayer monochromators, the excitation energy can be adjusted to 17.5 keV (molybdenum-K), 8.5 keV (tungsten-L) or 35 keV (tungsten-Brems). The detector is a Peltier-cooled energy-dispersive silicon drift detector (SDD) with an active area of 60 mm$^2$. Circular discs of a diameter of 30 mm and a thickness of 3 mm are used as sample carriers. A circular area with a diameter of approximately 5 mm in the centre of the sample carrier is the effective analysis area, which results from the superposition of the area excited by the X-ray beam and the field of view of the detector. Spectra of the samples were acquired for 1000 s using the molybdenum-K alpha excitation at a photon energy of 17.5 keV. By applying calibration samples as external standards, the ratio of fluorescence intensity to mass was calibrated for each element. Data analysis is performed using the commercial software Bruker T-ESPRIT (Version 1.0.1.443).

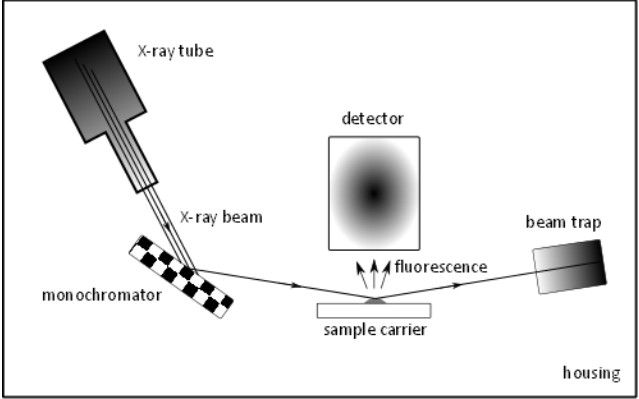

**Figure 4: Main components of the TXRF analysis setup. A polychromatic X-ray beam is monochromatized, impinges on the sample at a very flat angle, is totally reflected and ends in a beam trap; part of the X-ray radiation is absorbed in the sample and excites element-characteristic fluorescence, which is detected.**

Handling of the sample carriers was carried out in a clean room environment, especially after cleaning and during adhesive spin-coating or photographic documentation. The transport of the sample carriers was carried out in a dust-proof magazine. To prevent contamination during the TXRF analysis, the spectrometer housing was continuously purged with air filtered through a HEPA H14 filter (TROX GmbH, Neukirchen-Vluyn, Germany).





**4 Results and discussion**

**4.1 Impactor blank values**

Potential blank values due to material attrition are only to be expected for elements that the impactor is made of, which is mainly stainless steel, comprising the elements Fe, Cr and Ni essentially. Elements such as lead, arsenic and cadmium are only to be found in neglectable traces. The element fluorine, which is present in the sealing rings (about 65 wt.%), is not considered during analysis due to its low atomic number and the thereby resulting methodological insensitivity of TXRF. Consequently,

the main elements of stainless steel (Fe, Cr, Ni) are considered in the evaluation of possible blank values due to particle attrition of impactor material.

**Table 2. Fe, Cr, and Ni mean blank values of impactor stages 1 to 5 given in counts per 1000 s of the TXRF analysis and corresponding mass in ng, detected on the sample carriers after cleaning (TXRF no. 1), after assembly/disassembly five times (TXRF no. 2), after adhesive spin-coating (TXRF no. 3), and after sampling with particle-free air for 5 minutes (TXRF no 4).**

| | | TXRF no. 1 cleaned sample carriers | TXRF no. 2 5 x assembly/disassembly | TXRF no. 3 coated sample carriers | TXRF no. 4 sampling filtered air |
|---|---|---|---|---|---|
| Fe | fluorescence counts | 152.8 | 165.6 | 243.8 | 356.5 |
| | mass in ng | 0.004 | 0.005 | 0.007 | 0.010 |
| Cr | fluorescence counts | 93.2 | 84.8 | 84.,0 | 130 |
| | mass in ng | 0.005 | 0.004 | 0.004 | 0.006 |
| Ni | fluorescence counts | 71.2 | 78.0 | 98.0 | 91.8 |
| | mass in ng | 0.002 | 0.001 | 0.002 | 0.002 |


Table 2 shows the fluorescence intensity (counts) and the corresponding absolute mass of Fe, Cr and Ni as mean values of impactor stages 1 to 5 determined on the sample carriers of the blank value experiments. With the TXRF spectrometer applied, absolute lower detection limits of 0.005 ng for Fe and Cr, as well as 0.002 ng for Ni can be achieved in realistic conditions utilizing the sample carriers from the blank value experiment. Over all impactor stages, the absolute masses determined are in

the low picogram range and thus close to the absolute detection limit of the TXRF analysis.

With respect to Cr and Ni, 5 pg Cr and 2 pg Ni were detected for the cleaned sample carriers (TXRF analysis no.1), and no significant increase was observed when assembling and disassembling the impactor five times (TXRF analysis no.2), nor by coating the sample carriers with the adhesive (TXRF analysis no.3), nor by operating the impactor with filtered air (TXRF analysis no.4). For Fe, the signal shows a slight tendency to increase from 4 pg to 10 pg over the course of the experiment,

which, despite the cleanroom environment, could either be due to an introduction of iron from the air, or due to particles from impactor material attrition. In the case of particle attrition from the stainless-steel parts of the impactor, however, an increase in the nickel and chromium signal would be expected as well, due to stainless steel forming a protective layer of chromium dioxide on its surface. Therefore, it is not clear whether the observed slight increase is due to particle attrition or due to particles of external origin. Since the sample carriers were transported through normal indoor air in a dust proof packing during the

experiment, it cannot be excluded that the slight increase in iron is due to contamination during this transport. It is noticeable that the increase in iron (3 pg) was highest in the sampling of filtered air. Possibly, despite the high-efficiency filtered air, there was a small quantity of aerosol particles containing iron that passed through the high-efficiency filter and got collected within



the impactor. Overall, the very small variations in the measured masses of Fe, Cr and Ni in these experiments show that potential impactor blank values caused by attrition or sample carrier coating are extremely low and close to the TXRF detection
limit.

**4.2 Cross-contamination between subsequent sampling periods**

Since iron seems to be a highly abundant heavy metal in atmospheric air relative to other heavy metals (Seeger et al. 2021), iron was chosen to be analysed to assess potential cross-contamination between subsequent sampling periods. In Table 3, the counts and the corresponding absolute masses of iron are given which were determined during the cross-contamination
experiments. Photographs with perspective views of the four sample carriers can be found in Figure S2 (Supplementary Material).

**Table 3. Fe masses observed in cross-contamination experiments TXRF analyses no. 5 through no. 9**

|  | TXRF no. 5 cleaned and coated | TXRF no. 6 15-minute particle-free air clean impactor | TXRF no. 7 30-minute ambient air | TXRF no. 8 cleaned and coated | TXRF no. 9 15-minute particle-free air potentially contaminated impactor |
|---|---|---|---|---|---|
| sample carrier no. 9812 stage 4 (0.2 μm) | 415 counts 0.014 ng Fe | 439 counts 0.015 ng Fe | 38591 counts 1.321 ng Fe |  |  |
| sample carrier no. 9813 stage 3 (1 μm) | 389 counts 0.013 ng Fe | 395 counts 0.013 ng Fe | 173316 counts 5.933 ng Fe |  |  |
| sample carrier no. 9814 stage 4 (0.2 μm) |  |  |  | 387 counts 0.013 ng Fe | 431 counts 0.015 ng Fe |
| sample carrier no. 9815 stage 3 (1 μm) |  |  |  | 427 counts 0.014 ng Fe | 381 counts 0.013 ng Fe |

A fluorescence intensity of approximately 400 counts was typically determined for iron on cleaned and coated sample carriers
(TXRF no. 5). For example, with two cleaned and coated sample carriers (no. 9812 and no. 9813) installed in impactor stages 4 and 3, 415 counts and 389 counts, respectively, were determined. Considering the element-specific fluorescence sensitivity for iron, these counts correspond to an absolute mass of 0.014 ng and 0.013 ng, respectively. After installing these sample carriers into the impactor sampling filtered air for a 15-minute period at 5 slm, 439 counts and 395 counts were determined (TXRF no. 6). The very small increase of less than 24 counts and 6 counts, respectively, when sampling filtered air, compared
to cleaned sample carriers corresponds to a mass increase of less than 0.001 ng and is considered insignificant. Installing these sample carriers into the impactor again and collecting particles for 30 minutes at 5 slm from ambient air, 38591 counts (stage 4) and 173316 counts (stage 3) were determined (TXRF no. 7). These counts correspond to an absolute mass of 1.321 ng and 5.933 ng of iron, respectively. Particle mass concentrations of 13.55 μg/m$^3$ for the size fraction PM10, 8.93 μg/m$^3$ for the size fraction PM2.5, and 7.16 μg/m$^3$ for the size fraction PM1 were simultaneously determined by means of an optical aerosol
spectrometer. Accordingly, the impactor was regarded as "potentially particle contaminated" and was applied in this condition for the next experiment. Of two additional, cleaned and coated sample carriers (no. 9814 and no. 9815), 387 counts and 427 counts (TXRF no. 8) were determined for iron. After mounting these sample carriers into the impactor and sampling filtered



air for 15 minutes at 5 slm, 431 counts and 381 counts of iron fluorescence photons were determined (TXRF no. 9). These very small variations of less than 50 counts resulting from the operation with a "potentially particle contaminated" impactor is

considered to be insignificant and demonstrate that within the framework of these experiments cross-contamination is not observed for the impactor.

The masses of iron observed in the cross-contamination experiments are similar but slightly larger than the blank value of 10 pg of iron found in TXRF analysis no. 4. However, both the absolute fluorescence intensity of approximately 400 counts and the variation between both experiments are two to three orders of magnitude smaller compared to the increase in fluorescence

intensity when sampling ambient air for 30 minutes (TXRF analysis no. 7).

### 4.3 Particle collection in outdoor air

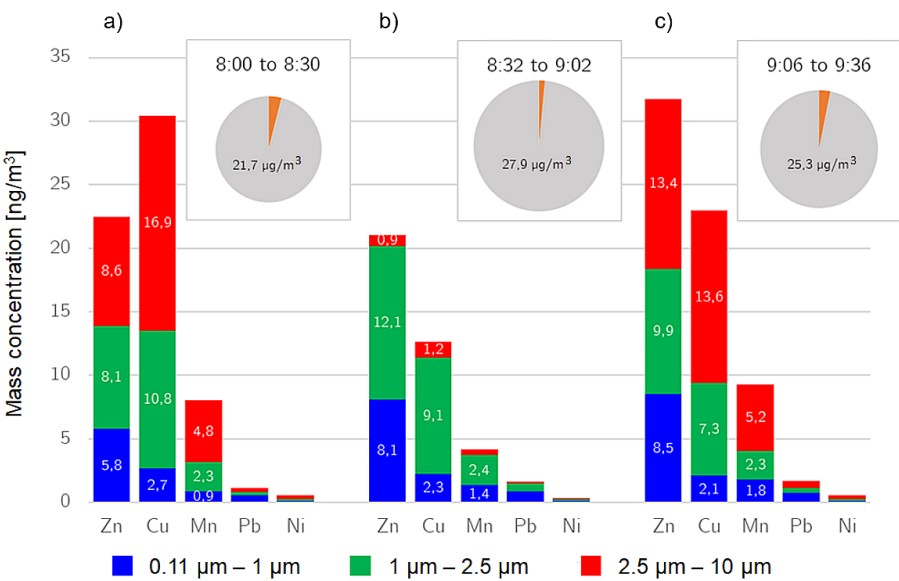

**Figure 5: Total PM10 (grey) and Fe (orange) concentrations, and trace metal concentrations (Zn, Cu, Mn, Pb, Ni) in three size fractions during three 30 min sampling periods (a, b, c) on 29 August 2022 at Potsdamer Platz, Berlin, Germany.**

To investigate the impactor in a real-world application, particles were sampled in ambient air with the impactor optimised for TXRF analysis in the morning of 29 August 2022 at Potsdamer Platz, Berlin, Germany. Airborne particles were collected during three consecutive 30 min sampling periods from 08:00 to 08:30, 08:32 to 09:02 and 09:06 to 09:36. The bar chart in Figure 5 shows the mass concentrations of the trace metals Zn, Cu, Mn, Pb and Ni collected during these three sampling periods in three size fractions, i.e. PM1 (top of blue bar), PM2.5 (top of green bar) and PM10 (top of red bar). In addition, the

pie charts give the corresponding PM10 concentrations estimated from an Fidas Frog optical aerosol spectrometer measurement, and the Fe mass fractions in PM10 (orange). The PM10 concentrations varied from 21.7 µg m⁻³ in the first 30 min sampling period to 27.9 µg m⁻³ in the second 30 min sampling period. The contribution of Fe to PM10 varied from 3.7 % (0.81 µg m⁻³) in the first period, 1.4 % (0.38 µg m⁻³) in the second period to 2.9 % (0.73 µg m⁻³) in the third period. Zn and Cu concentrations vary from 12.6 to 31.8 ng m⁻³ in PM10, and Mn, Pb and Ni concentrations are below 10 ng m⁻³ in PM10.

Arsenic could not be detected in the samples. The observed 30 min concentrations of Pb in PM10 (ranging from 1.1 to 1.7 ng m⁻³) and Ni in PM10 (ranging from 0.4 to 0.6 ng m⁻³) are lower but consistent with the annual mean concentrations of Pb in PM10 (4.4 ng m⁻³ in 2021) and Ni in PM10 (0.6 ng m⁻³ in 2021) measured at the Berlin air quality network BLUME station Berlin-Neukölln (Berliner Luftgütemessnetz 2023). Detailed results of heavy metal concentrations in PM10, PM2.5 and PM1





can be found in Table S2 (Supplementary Material). It should be kept in mind that the separation diameter of stage 5 is approximately 0.095 µm and thus, particles with a smaller aerodynamic diameter were not fully captured in this impactor configuration. Although the mass contribution of these ultrafine particles can be considered very small (e.g. Seeger et al. 2021), additional nozzle modules with smaller separation diameters can be added to extend the lower size range of the impactor if required.

In the present case study, it is interesting to note that considerable concentrations of trace metals can be found in the PM2.5 fraction and even in the PM1 fraction (especially Zn and Pb). In the second sampling period (Figure 5b), the concentrations of Fe and other trace metals in the coarse mode from 2.5 µm to 10 µm are much lower than in the first and third sampling period. This indicates a transient change in the chemical composition of coarse mode particles, while variations in the fine fraction are less pronounced. Such short-term variations, possibly due to variable source strength patterns, can only be observed with measurements of sufficiently high time resolution. It is evident from Figure 5 that with the optimised impactor and the high sensitivity of TXRF analysis, 30 min sampling times are sufficient to reliably quantify trace metal concentrations including lead and nickel in PM10, PM2.5 and PM1 size fractions at moderate atmospheric concentrations.

## 5 Conclusions and outlook

The newly developed impactor optimised for TXRF analysis opens new perspectives in the determination and quantification of particulate trace metals and other elements with high time resolution. Due to the compact arrangement of the impactor nozzles, the high detection sensitivity of the TXRF analysis method can be fully exploited. We demonstrate that a large number of different heavy metals can be detected and quantified in the PM10, PM2.5 and PM1 size fractions after collection periods of 30 minutes. Thus, snapshot sampling with collection periods of significantly less than one hour is possible. Sampling carriers can be exchanged in less than five minutes including disassembling and assembling of the impactor stages, and the total procedure from sampling with the impactor to the result of the TXRF elemental analysis takes only a few hours. This offers new possibilities for the identification of pollutant sources and the evaluation of protective measures. The flow rate required for sampling is relatively low at 5 slm, which means that the associated pump can be operated with a portable rechargeable energy source. This enables flexible, mobile sample collection, even in public spaces with a high volume of people or traffic. Blank values and cross-contamination potential as previously reported for stainless steel impactors were not detected with the new impactor, even with the low detection limits of TXRF analysis. Furthermore, the new spin-coating method provides adhesive coating of the sample carriers with very low surface roughness, which is advantageous for TXRF analysis.

A characterisation of the separation efficiency of individual impactor stages is currently in preparation. Calibration of the classification diameters and efficiency curves of individual impactor stages is possible through an adjusted design of the impactor nozzle arrangements in the exchangeable nozzle modules. A reduced nozzle diameter and an increased number of nozzles, for example by laser drilling, would reduce the differential pressure across the nozzle stage and facilitate the addition of impactor stages with smaller separation diameters. Care should be taken to adjust the sampling time according to the expected atmospheric particle burden. Overloading of the sample carriers with particles will lead to self-absorption of fluorescence radiation within the sample matrix. In this case, applying a linear relationship between fluorescence intensity and element mass concentration will underestimate the actual concentration. Further improvement of quantitative TRXF analysis is possible with approaches such as grazing incidence-x-ray fluorescence (e.g. Hönicke et al. 2019), where variation of the incident angle of the excitation beam yields fluorescence from different parts of the deposited sample.





**Author Contributions**

Formal analysis, Investigation, Methodology, Visualization and Writing – original draft preparation by CC. Conceptualization, Validation and Writing – review & editing by CC and AH.

**Competing interests**

The authors declare no competing interests.

**Acknowledgements**

The authors are grateful for funding by Bruker Nano GmbH. Additional technical support by the Bruker Nano staff is gratefully acknowledged.



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
