# Peer review of "Development of a cascade impactor optimised for size-fractionated analysis of aerosol metal content by total reflection X-ray fluorescence spectroscopy (TXRF)"

_Atmospheric Measurement Techniques, 2023_

## Author Comment (AC1)

**Response to comments of Referee #1**

We would like to thank Referee #1 for his valuable and thoughtful comments, which helped us to improve the content and quality of our manuscript. In the following we have addressed all the comments of the Referee #1 and incorporated changes in the manuscript as follows:

Blue: Comments of the Referee

Black: Answers of Authors

*Black, italic, "": "Changes in the manuscript"*

General:

1. Does the paper address relevant scientific questions within the scope of AMT? Yes
2. Does the paper present novel concepts, ideas, tools, or data? Yes
3. Are substantial conclusions reached? Partly, see major points of criticism.
4. Are the scientific methods and assumptions valid and clearly outlined? Partly, see major points of criticism.
5. Are the results sufficient to support the interpretations and conclusions? Partly, see major points of criticism.
6. Is the description of experiments and calculations sufficiently complete and precise to allow their reproduction by fellow scientists (traceability of results)? Partly, see major points of criticism.
7. Do the authors give proper credit to related work and clearly indicate their own new/original contribution? Yes
8. Does the title clearly reflect the contents of the paper? Yes
9. Does the abstract provide a concise and complete summary? Yes
10. Is the overall presentation well-structured and clear? Yes
11. Is the language fluent and precise? Yes
12. Are mathematical formulae, symbols, abbreviations, and units correctly defined and used? Overall, yes
13. Should any parts of the paper (text, formulae, figures, tables) be clarified, reduced, combined, or eliminated? Some parts need clarification and additional detail information. See below.
14. Are the number and quality of references appropriate? Yes
15. Is the amount and quality of supplementary material appropriate? Yes

Thank you very much for this assessment.

1.1. Line 55 to 59 describes the technical challenge to which the paper is dedicated: Increasing the detection efficiency of TXRF by complete excitation of the deposit and complete detection of the fluorescence response. The authors present a solution by shrinking the lateral diameters of centric deposit patterns on cascade substrates (carriers) to less than 5 mm and describe an appropriate cascade impactor design. To

the reader the 5 mm criterion seems to be an arbitrary choice and a justification is lacking. A justification could e. g. be derived from an excitation beam profile analysis along with a description of the detector's aperture and lateral efficiency distribution. Both would, e. g. for the used Bruker TStar spectrometer, clearly reveal that other than the authors implicitly assume even within the "5 mm area" the excitation and detection would have radial dependencies, i.e., decrease with distance to the center.

In the TXRF spectrometer used in this study, the excitation beam covers a rectangular area on the sample carrier with a width of 6 mm and a length of 30 mm. This means that the excitation beam covers the area of the sample carrier completely in which the new impactor deposits the particles, namely the centric circle with a diameter of less than 5 mm. This information was added to the revised manuscript in section 3.3, lines 274 to 277 of the revised manuscript, to justify the targeted size of the deposition pattern ("less than 5 mm") and to make it more comprehensible:

> „A circular area with a diameter of approximately 5 mm in the centre of the sample carrier is the effective analysis area, which results from the superposition of the area excited by the X-ray beam, _namely a rectangular area with a width of 6 mm and a length of 30 mm,_ and the field of view of the detector."

The authors agree with the referee that both excitation and detection are not homogeneous over the entire surface of the deposited particles.
These inhomogeneities of the TXRF spectrometer are specific for the model and/or manufacturer of the TXRF spectrometer applied for analysis, and therefore these inhomogeneities are not compensated by the design of the new impactor.
A corresponding statement was added to the revised manuscript in section 3.3, line 277 to 279 of the revised manuscript for clarification:

> _„Both excitation and detection are not homogeneous over the entire surface of the deposited particles. These inhomogeneities of the TXRF spectrometer are specific for the model of the TXRF spectrometer applied for analysis, and therefore these inhomogeneities are not compensated by the design of the new impactor."_

Last but not least, we would like to reiterate as pointed out in the manuscript that the optimisation of the new impactor is not only in a simple shrinking of the pattern of the deposited particles, but also, for example, in an optimised lateral arrangement of the nozzles in order to enable an optimised cross-flow out of the impaction area. In addition, the impactor was designed for a comparatively low flow rate of 5 l/min in order to reduce the requirements for the pump and its power source; this significantly improves the possibilities for mobile and portable use of the new impactor.

1.2. In line 120 authors describe a Reynolds number of ~3000 in the nozzles as a design criterion. This is in my understanding for cylindrical pipes the transition regime between turbulent (>3000) and laminar (<=2300) flow. In line 125 to 130 authors describe that the flow profile in the nozzles should be "plug-shaped", i. e. turbulent, rather than "parabolic" aka laminar. This is in contradiction to line 137, where a Reynolds number of < 3000 is described as desirable. The relation between number and diameter of nozzles and mean velocity could be described more precisely and as a

design challenge. It is desirable to have the finally applied ratios of nozzle lengths to diameters presented as a result of theoretical calculations or CFD modelling.

Both (a) an upper limit $Re_{max}$ = 3000 for the Reynolds number $Re$ and (b) a laminar rather than a turbulent nozzle flow are design criteria in full accordance with VDI Guideline 2066, Part 10, which specifies (in point 4.2.c) that the Reynolds number for the nozzle flow in the impactor should be within a range of laminar flow between 100 and 3000.

At the same time, a plug-shaped flow profile at the end of the nozzles is preferable over a parabolic flow profile, as the differences between the flow velocity in the centre of the nozzle and the flow velocity at the edge of the nozzle are smaller with a plug-shaped flow profile, which is advantageous for steep separation characteristics. Since a parabolic flow profile is formed in a pipe in laminar flow conditions only after some distance from the pipe inlet, the length $l_n$ of the cylindrical section of the nozzles was chosen to be small compared to the diameter $d_n$ of the cylindrical section of the nozzles in order to avoid the complete formation of a parabolic flow profile and maintain a plug-shaped flow profile. A corresponding statement was added to the revised manuscript in section 2.2, line 133 to 136 of the revised manuscript for clarification:

> „*Since a parabolic flow profile is formed in a pipe in laminar flow conditions only after some distance from the pipe inlet, Ll̶ong classifying nozzles favour the development of a parabolic, pipe-like flow profile, resulting in a less steep separation curve. Therefore, the cylindrical throat length of the classifying nozzles $l_n$ was kept short in order to facilitate the formation of a plug-shaped flow profile.*"

The finally applied ratios $l_n/d_n$ are between 0.5 and 1.5, as described in line 130 of the preprint manuscript.

The relations between the separation diameter $d_{ae50}$, the number $N_n$ and diameter $d_n$ of the nozzles, the mean velocity $v_0$, and the volumetric flow rate $Q$, which are the relevant parameters for the design of the impactor nozzles, are given by equations 1, 2, and 3 in chapter 2.2, lines 106 to 125 of the preprint manuscript. The applied values are summarized in Table 1, line 135 of the preprint manuscript.

1.3. Attrition (section 3.3.1). To me, sample contamination by metallic attrition appears a bit far-fetched. This section could possibly be eliminated. Authors could e. g. have made a point in discussing attrition by solid, abrasive particles. The authors also did not consider the possible contamination of a quartz disc by unintended mechanical contact with metallic parts (e. g. impactor structure, tweezers…) or other materials which have had mechanical contact to metals. This deems to me a more likely possible source of contamination. The presentation of related results in section 4, Table 2 has some deficiencies as will be discussed below.

The authors respect the referee's doubts about the relevance of metallic attrition, but respectfully prefer to keep this section in the manuscript.

In a previous study, Klockenkämper et al. (1995) determined blank values of 40 % on average for stainless steel impactors in relation to the actual measured values, and explicitly attributed these high blank values to abrasion of metallic particles from the impactor. As a result, Klockenkämper et al. (1995) recommend avoiding stainless steel as a material for an impactor. Since we did not find any literature that refuted this contamination potential, we considered it necessary to carry out corresponding investigations ourselves.

We agree with the referee that the possible contamination of the sample carrier by unintended mechanical contact with metallic parts is a relevant potential source of contamination. However, this potential source is not due to the impactor material or design but rather due to the handling of samples.

In view of referee's comment, we introduced a corresponding explicit advice in section 3.1.1, line 203 to 205 of the revised manuscript:
> „…of the above-mentioned composition. _In addition, non-metallic tweezers were used to manipulate the sample carriers, and even beyond that, undesirable metallic contamination of the sample carriers was avoided by careful handling._ To determine the…"

Second, as far as the impactor structure is concerned, numerous design measures were described in the manuscript to prevent metallic contamination of the sample carrier. For example, the sample carrier is fixed in the impactor by means of a (non-metallic) "elastic mounting ring made from laser sintered polyamide", see lines 90 and 91 of the preprint manuscript, in order to avoid contact between the surface of the sample carrier examined by TXRF analysis and metallic parts.

In view of referee's comment, we added a clarification to the caption of Figure 1 in line 100/101 of the revised manuscript:
> "…accommodating the nozzle module (red and green)_, a non-metallic mounting ring (olive) for fixing the sample carrier,_ and a lower body (light grey) accommodating…"

**1.4. Line 265: The statement on the "effective area" is vague.**

As already indicated in item 1.1, the authors have amended the manuscript to clarify why a circular area with a diameter of approximately 5 mm in the centre of the sample carrier is regarded as "the effective analysis area, which results from the superposition of the area excited by the X-ray beam, namely a rectangular area with a width of 6 mm and a length of 30 mm, and the field of view of the detector of the TXRF spectrometer used in this study."

As pointed out above, the efficiency will most probably have a radial dependency. Authors do not present an argument why this could be ignored.

It is not the intention of the authors to evoke the impression that radial

inhomogeneities of the excitation beam and/or the detection of fluorescence of the TXRF spectrometer should be ignored in the TXRF analysis. However, when developing the new impactor, the aim was to ensure that the aerosol particles are deposited on the sample carrier in an area that is, on the one hand, completely covered by the excitation beam and, on the other hand, completely covered by the field of view of the detector. It is true, as the referee points out, those radial inhomogeneities in the TXRF analysis affect the result. Therefore, as already mentioned in item 1.1 above, a corresponding statement has been added to the revised manuscript in section 3.3, line 274 to 277 for clarification.

Line 269: "Calibration samples as external standards". This is a crucial point in quantitative TXRF analysis, and the paper is much too vague here. The authors should explain the calibration samples in detail. I assume that these were prepared by drying a small drop of an aqueous solution with known element quantity (probably Yttrium?) in the center of the carrier.

The authors have amended the manuscript in section 3.3, line 281 to 284 of the revised manuscript to provide more clarity to the calibration process:

> "… mass was calibrated for each element. *For this purpose, the ratio (counts per mass unit) of fluorescence intensity (TXRF measurement) and mass was first determined for reference samples of which the mass per element was known. Subsequently, the samples with the impacted aerosol particles were quantified with the ratio determined in this way.*"

I have the impression that the authors implicitly assume for the whole deposition pattern area the same detection efficiency as for a punctiform centrical deposition of an internal standard and they should give a proper justification for this assumption. As the commercial Bruker software for the TStar offers several options for quantification the chosen method (internal fundamental parameters, internal standard…) should be mentioned.

The authors don't want to give the impression described by the referee. Indeed, it can make a difference for TXRF analysis whether the sample to be analysed is presented in a single closed surface, preferably still centred on the sample carrier, or as a multi-point deposition pattern, as is the case when using an impactor. This is discussed, among others, in the reference Prost et al. (2017) cited in the original manuscript. It is beyond the scope of the manuscript to develop a TXRF calibration strategy for impaction deposition patterns. The objective is the development of a new impactor that achieves improved results with the TXRF spectrometers available on the market, in particular through the detection of not only a part but of all aerosol particles impacted on the sample carrier.

1.5. Chapter 4, Table 2: "…mean blank values of impactor stages 1 to 5 …": I guess that this are the means over repeated measurements. Lines 291 to 295. I would expect a proper measure of variance, at least an estimation of the uncertainty. Each single

measurement should come with an uncertainty provided by the analyzing software. The absolute lower detection limits (LLOD) should be quantified. Line 299 to 301: The significance of the statement should be reevaluated considering realistic uncertainties in the measurements. I am reluctant to accept conclusions without proper consideration of uncertainties, the more so as the data are close to the LLOD.

The authors agree with the referee that the blank values measured and summarized in Table 2 of section 4.1 are very small and very close to the absolute lower detection limits given in line 293 of the preprint manuscript. This is ultimately due to the fact that the sample carriers were not contaminated in the course of the experiment. The mean values are calculated from five measurements, respectively. The maximal standard deviation was 0.0033 ng for Fe, 0.0016 ng for Cr, and 0.0008 ng for Ni. A corresponding statement has been added in section 4.1, line 307 to 309 of the revised manuscript:

> "…of the blank value experiments. _The mean values are calculated from five measurements, respectively. The maximal standard deviation was 0.0033 ng for Fe, 0.0016 ng for Cr, and 0.0008 ng for Ni._ With the TXRF spectrometer applied…"

Therefore, the statement in lines 299 to 301 of the preprint manuscript concerning the observed increase of the mass of Fe from 0.004 ng to 0.010 ng over the course of the experiment seems to be reasonable, whereas for the elements Cr and Ni the measured variations remain within the standard deviation.

1.6.  Chapter 4 from line 345 on and Figure 5 and Supplement Table S.2: "Figure 5 shows the mass concentrations of the trace metals Zn, Cu, Mn, Pb and Ni collected during these three sampling periods in three size fractions, i. e. PM1 (top of blue bar), PM2.5 (top of green bar) and PM10 (top of red bar).". Authors should describe how they calculated the PMx mass fractions from the masses determined by TXRF on the five cascade impactor stages.

The authors are pleased to follow up on this recommendation and amended line 366 to 368 of the revised manuscript as follows:

> "…periods in three size fractions, i.e. PM1 (top of blue bar_, i.e. sum of masses of impactor stages 4 and 5_), PM2.5 (top of green bar_, i.e. sum of masses of impactor stages 3, 4, and 5_) and PM10 (top of red bar_, i.e. sum of masses of impactor stages 2, 3, 4 and 5_)."

Were for the 0.11 to 1 µm fraction the masses from stages 3 to 5 just summed up, and accordingly for the other PM fractions in Figure 5?

Stage 3 has a separation diameter of 0.915 µm (or approximately 1 µm). This means that – in an idealised view – particles with an aerodynamic diameter of 1 µm are impacted on stage 3 with 50 % efficiency and larger particles more efficiently. Therefore, the mass of the particles impacted on stage 3 must not be attributed to the PM1 fraction, but the fraction of PM1 is to be calculated by summing the masses of the particles impacted on stages 4 and 5. Other PM fractions in Figure 5

were determined accordingly, which is now clarified in the revised manuscript.

If the coarse (10 µm) stage was included into the calculations at least all values labeled as PM10 (top of red bar) do contain a priori unknown quantities of particles larger than 10 µm as no preseparator was reportedly used and the separation curve steepness according to DIN 481 or US EPA was not determined. However, the photo (Fig. 3) probably shows an inlet separator which is not described at all.

Stage 1 has a separation diameter of 9.96 µm (or approximately 10 µm).
This means that – in an idealised view – particles with an aerodynamic diameter of 10 µm are impacted on stage 1 with 50 % efficiency and larger particles more efficiently. Thus, stage 1 acts as a pre-separator for particles larger than 10 µm. Accordingly, the mass of the particles impacted on stage 1 must not be attributed to the PM10 fraction, but the fraction PM10 is to be calculated by summing the masses of the particles impacted on stages 2, 3, 4 and 5.

As stated above, the calculation scheme is now explicitly stated in the amended manuscript in order to avoid misunderstandings.

1.7.  Conclusions and Outlook seems not well-balanced: One half of the text can be considered merely as outlook.

The first part comprising 13 lines (lines 373 to 386 of the preprint manuscript) summarises what has been achieved with the newly developed impactor. The second part, comprising 9 lines (lines 387 to 396 of the preprint manuscript), gives some concrete recommendations as to which measures (e.g. laser drilling of the nozzles) can be used to achieve further improvements or extensions (e.g. even smaller separation diameters), or which problems have to be avoided (no particle overload of the sample carriers). We consider both parts relevant and would prefer to keep this section as is.

The following three points can in essence be regarded original results from this work:
1.  "Due to the compact arrangement of the impactor 375 nozzles, the high detection sensitivity of the TXRF analysis method can be fully exploited." A convincing proof of this statement is however lacking. The statement in lines 375 to 379 is valid.
2.  The statement on the "new spin-coating method" is valid.
3.  No contamination issues during mounting/dismounting of carriers.

As described in the manuscript, the deposition pattern of impactors available so far, especially in the impactor stages for the sub-µm particles, have such large lateral dimensions that a substantial part of the impacted particles is deposited outside the area covered by the excitation beam and/or that the fluorescence of a substantial part of the impacted particles could not be captured by the detector. All these not-excited or not-detectable particles were a priori lost to TXRF analysis. As a result, the high potential of TXRF analysis, especially in terms of small sample air flows
(-> small pumps, low energy consumption, low weight, mobile application, no local

particle overload, etc.) and short sampling time (-> high temporal resolution, thus possibility of identification of particle sources), cannot be fully exploited.

Due to the compact arrangement of the nozzles of the newly developed impactor any deposition of particles outside the excitation beam and outside the detectable area is omitted. No particle gets lost for the TXRF analysis. Instead, all particles impacted on the sample carrier are covered by the excitation beam, and the fluorescence of all particles impacted on the sample carrier is captured by the detector.

Three particle collections in outdoor air were conducted with the new impactor, each for 30 minutes within a total of only 96 minutes, with fully mobile equipment including battery powered pumps. The subsequent TXRF analysis reveals metal concentrations very close to the values published by governmental measuring agencies.

In view of referee's comment the authors have amended section 5, line 393 to 395 of the revised manuscript in order to clarify the contribution of the compact arrangement of the impact nozzles to the detection sensitivity of the TXRF analysis:

> …Due to the compact arrangement of the impactor nozzles, the entire sample interacts with the TXRF excitation beam and contributes to the TXRF analysis signal, thereby supporting the high detection sensitivity of the TXRF analysis method .

**Minor points / editorial suggestions:**

Line 76 to 77: The citation refers to rather broad outdoor aerosol size distributions, e. g. accumulation or coarse mode. There might be outdoor scenarios where the Aitken mode is interesting to observe even with high time resolution, or indoor scenarios were the < 0.1 µm fraction is dominant. Authors should comment on that.

The authors thank the referee for this advice and have amended lines 77 to 84 of the revised manuscript as follows:

> "*Previous TXRF analysis results of impactor samples (e.g. Seeger et al., 2021) suggest that particles of 0.1 µm diameter or less make only a minor contribution to particulate mass assuming a typical outdoor aerosol size distribution with significant contributions of the accumulation mode and coarse modes. Therefore, a fourth stage with a separation diameter of approximately 0.1 µm can be used to collect almost the entire PM1 fraction. In the present study, we apply a fourth stage with a nominal separation diameter of 0.13 µm, and a fifth stage with a nominal separation diameter of 0.095 µm for experimental purposes. In special circumstances it might be important to collect ultrafine particles, e.g. with an additional impactor stage with a separation diameter of e.g. 0.02 or 0.01 µm, as outlined in section 5 of the manuscript, or alternatively a final filter stage.*"

Line 82: "slm" (standard litre per minute) refers to us-american nomenclature with standard

conditions 0 deg Celsius (32 deg F) and 1.0125 bar (14.69 psia). European nomenclature uses subscript "$_s$" for standard flow at 20 C and 1.0125 bar, e.g., $m^3_s$/h. Standard flow at 0 C and 1.0125 bar is in Europe expressed as e.g., $m^3_n$ / h with subscript "$_n$"

For reasons of clarity, the authors prefer to keep the unit "slm" for the mass flow rate: The unit "slm" is defined in the manuscript as "standard litres per minute"; the referenced standard conditions (1.013 hPa and 20 °C) are also explicitly specified in the manuscript, see section 2.1, line 82 of the preprint manuscript. The unit "slm" is therefore clearly defined in the present manuscript.

Line 165: "in the low single-digit percentage range" ?? "few percent relative deviation" sounds better.

The authors thank the referee for this advice and have amended line 172 of the revised manuscript accordingly:

"...resulted in relative deviations of only a few percent in relation to from the respective nominal diameters in the low single-digit percentage range..."

---

## Author Comment (AC2)

**Response to comments of Referee #2**

We would like to thank Referee #2 for his valuable and thoughtful comments, which helped us to improve the content and quality of our manuscript. In the following we have addressed all the comments of the Referee #2 and incorporated changes in the manuscript as follows:

Blue: Comments of the Referee

Black: Answers of Authors

*Black, italic, "": "Changes in the manuscript"*

General:

This work by Crazzolara and Held developed a new cascade impactor that improves the detection limits of heavy metals in aerosol particles. What's more, the newly equipment was in small size with low detection limits for metal elements, which is beneficial to monitoring work in field observation. This article is well organized, informative and in line with the scope of AMT. I suggested that the manuscript can be accepted and published after addressing the following concerns.

Thank you very much for this assessment.

2.1.  The author stated that the detection limit of TXRF is superior to XRF (Yoneda and Horiuchi, 1971), and can reach down to a few picograms of absolute mass on the sample carrier substrate (Streli 2006). Can the author give the detailed detection limits of TXRF of some elements for comparison? And more latest references should be provided here.

The detection limits achievable by TXRF analysis depend on several factors, in particular the type of fluorescence excitation, the respective sample characteristics, and the TXRF spectrometer used. For the configuration used in the present study (X-ray excitation, $SiO_2$ sample carrier, aerosol particles deposited on the sample carrier by an impactor, portable Bruker S4 T-Star spectrometer), the lower detection limits are, for example, 0.005 ng for Fe and Cr and 0.002 ng for Ni. Recent publications report that even detection limits in the range of fg ($10^{-15}$ g) can be achieved with TXRF analysis, for example by using synchrotron excitation of the sample.

The manuscript has been revised in section 1, line 52 to 55 of the revised manuscript by adding the corresponding references:

> *"… and can reach down to a few picograms of absolute mass on the sample carrier substrate (Streli 2006). Even detection limits in the range of fg ($10^{-15}$ g) can be achieved with TXRF analysis (Eichert 2020), and also light elements (with low Z-number) can be excited effectively by using synchrotron excitation of the sample (Beckhoff et al. 2007, Streli et al. 2008). Recently…"*

The following references have been added to the preprint manuscript:

*Eichert, D. (2020). The Fundamentals of Total Reflection X-ray Fluorescence. Spectroscopy, 35(8), 20-24.*

*Beckhoff, B., Fliegauf, R., Kolbe, M., Müller, M., Weser, J., & Ulm, G. (2007). Reference-free total reflection X-ray fluorescence analysis of semiconductor surfaces with synchrotron radiation. Analytical chemistry, 79(20), 7873-7882.*

*Streli, C., Wobrauschek, P., Meirer, F., & Pepponi, G. (2008). Synchrotron radiation induced TXRF. Journal of Analytical Atomic Spectrometry, 23(6), 792-798.*

2.2. Line 84-85: "…… without exceeding a critical Reynolds number of 3000" and Table 1. It puzzles me that why the criterion set as 3000 in a circular area with a diameter of less than 5 mm on stages 3, 4 and 5?

A laminar flow (which is generally desirable in the impactor nozzles) can turn into a turbulent flow if the Reynolds number $Re$ to be calculated according to equation 3 in the manuscript exceeds a critical value. For this reason, a critical Reynolds number was set as an upper limit when designing the new impactor.

The VDI 2066 guideline on $PM_{10}$ and $PM_{2,5}$ particulate matter measurement by impaction method specifies in Part 10, point 4.2.c) that the Reynolds number for the nozzle flow in the impactor should be within a range of laminar flow between 100 and 3000. Accordingly, when designing the new impactor, the number and diameter of the impactor nozzles in stages 3 to 5 were selected so that the Reynolds number does not exceed 3000.

And why small nominal nozzle diameters corresponded to the high Reynolds number?

The Reynolds number $Re$ is calculated for the air flow in the impactor nozzles according to equation 3 in the manuscript. Accordingly, the Reynolds number $Re$ is proportional to both the flow velocity $v_o$ and the nozzle diameter $d_n$. The following rough calculation shows that smaller nozzle diameters result in higher Reynolds numbers: If, for example, the nozzle diameter $d_n$ is reduced by 50 %, this leads to a reduction of the nozzle cross-sectional area to 25 %; therefore, assuming a constant gas volume flow, the flow velocity $v_o$ increases to 400 %. As a result, the Reynolds number doubles when the nozzle diameter is halved.

2.3. Line 243-245: The operating temperature of the sensor is ranged from -20 to +80 °C. The question is whether the sensor have an applicable relative humidity RH range? For example, in the coastal regions, high humidity and salinity environment may cause damage to the instrument, such as corrosion.

Due to its measuring principle, the sensor element is heated during operation so that condensation is not to be expected under normal atmospheric conditions.

For long-term exposure, the mass flow sensor is specified for a maximum humidity at a dew point of 40 °C (100 % relative humidity at 40 °C), which corresponds to an absolute humidity of more than 50 g/m$^3$. Furthermore, the mass flow sensor is arranged downstream of the impactor, and in addition, a filter element is arranged upstream of the sensor element of the mass flow sensor so that no contamination of the sensor element with salt particles can occur. In addition, the sensitive element of the sensor is passivated with silicon nitride.

The authors would also like to thank the referee for this comment, as it drew their attention to an error in the manuscript. The temperature range presented in line 244 of the preprint manuscript is for a different mass flow sensor of the same manufacturer; the correct operation temperature range for the mass flow sensor SFM4300-20-P is only +5 to +50 °C. Although this does not affect the measurement results presented in the manuscript, the manuscript was corrected in section 3.2, line 252/253 of the revised manuscript accordingly:

> "*The sensor is factory-calibrated, has an operating temperature range of  +5 to +50 °C and provides a temperature-compensated output signal.*"

2.4. Line 251-254: How to set the duration of the sample time to ensure that the sample meets the needs of the analysis? For example, if the collection time is too short in a very clean environment, the sample volume may be not sufficient for the test, but if the collection time is too long, the sample may be overloaded.

This comment by the referee applies to impactor sampling in general. In practice, expected concentrations of the chemical elements of interest can often be estimated from previous studies. This information can be used to calculate the expected optimum sampling time.

2.5. Line 295: The detection limits of Fe, Cr and Ni by TXRF analysis should be given here.

The detection limits are 0.005 ng for Fe and Cr, as well as 0.002 ng for Ni, as indicated in section 4.1, line 309/310 of the revised manuscript. The wording of this sentence has been amended to enhance clarity and avoid misunderstanding:

> "*…absolute lower detection limits of 0.005 ng for Fe and Cr, as well as 0.002 ng for Ni  were achieved in realistic conditions utilizing the sample carriers from the blank value experiment.*"

2.6. In the section "4.3 Collection of particles in outdoor air", does the lower mass concentrations of Pb in PM10 (ranging from 1.1 to 1.7 ng m-3) and Ni in PM10 (ranging from 0.4 to 0.6 ng m-3) imply that a sampling period of 30 min is not sufficient for analysis.

No, the reported lower mass concentrations of Pb and Ni are well above the detection

limits. The detection limit for Pb was 0.001 ng or 1 pg absolute mass. As the volume of the sampled air was 150 litres at standard condition (30 minutes at 5 slm), the lowest measured concentration of 1.1 ng/m$^3$ Pb corresponds to an absolute particle mass of 0,165 ng or 165 pg Pb and is therefore well above the detection limit.

The detection limit for Ni was 0.005 ng or 5 pg absolute mass. Accordingly, the lowest measured concentration of 0.4 ng/m$^3$ Ni corresponds to an absolute particle mass of 60 ng Pb and is also well above the detection limit.

2.7. Have the authors compared the capture efficiency of the new cascade impactor with other commercially cascade impactors such as MOUDI Impactor Series (TSI Incorporated, USA) or Andersen Cascade Impactor (Tisch Environmental, Inc. USA)?

No, a direct comparison with other commercially available impactors has not been performed. A quantitative comparison of chemical composition using TXRF would be difficult because the deposition patterns of these impactors have lateral dimensions in some stages that lie outside the excitation range and/or the detection range of the TXRF spectrometer used in the present study. However, the mass concentrations determined with the new impactor and shown in section 4.3 are consistent with the annual mean concentrations of Pb in PM10 (4.4 ng/m$^3$ in 2021) and Ni in PM10 (0.6 ng/m$^3$ in 2021) measured at the Berlin air quality network BLUME station Berlin-Neukölln.

Minor concerns:

1. PM10, PM5 and PM1 should be subscripted. Many similar issues in the manuscript.

The authors would like to thank the referee for pointing this out and adapted this throughout the manuscript:

    "PM10"    ->    "PM$_{10}$"
    "PM2.5"  ->    "PM$_{2.5}$"
    "PM1"    ->    "PM$_1$"

---

## Author Comment (AC3)

**Response to comments of Referee #3**

We would like to thank Referee #3 for his valuable comments, which helped us to improve the content and quality of our manuscript. In the following we have addressed all the comments of the Referee #3 and incorporated changes in the manuscript as follows:

Blue: Comments of the Referee

Black: Answers of Authors

*Black, italic, "": "Changes in the manuscript"*

General:

In this study, a new impactor optimised for TXRF analysis was developed, demonstrating that a large number of different heavy metals can be detected and quantified in the PM10, PM2.5 and PM1 size fractions after collection periods of 30 minutes. Overall, the new impactor bears potential to improve the quantification of particulate trace metals and other elements in PM10, PM2.5 and PM1 with high time resolution. Major comments are as follows:

Thank you very much for this assessment.

3.1   In the introduction, it is said that "Despite promising results, commercial impactors are not fully optimised for TXRF analysis: The area on the sample carrier in which the classifying nozzles deposit the particles is usually significantly larger than the area analysed by TXRF." The inappropriate area was the only limitation? More advances should be showed.

The overall design of the newly developed impactor has multiple benefits:
By arranging the impactor nozzles of the newly developed impactor in such a way that all particles impacted on the respective sample carrier can contribute to the TXRF analysis (and not losing a significant proportion of the impacted particles for the TXRF analysis right from the outset), shorter sampling times are possible, as explained in lines 61 and 62 (introduction) of the preprint manuscript. This opens up new possibilities for identifying pollutant sources, as outlined in line 380 of the preprint manuscript.

The reduction of the lateral dimensions of the deposition patterns or the corresponding arrangement of the impactor nozzles could not be achieved simply by compressing a previous deposition pattern, but rather the number and lateral arrangement of the impactor nozzles had to be recalculated as well as the diameters of the impactor nozzles, as described in detail in section 2.2 of the preprint manuscript.

As a result of these considerations, the newly developed impactor was designed for a significantly reduced gas mass flow compared to commercially available impactors, which facilitates the use of smaller pumps, and thus, portable and mobile battery-powered operation of the impactor in the field, as outlined in lines 381 to 384 of the preprint manuscript.

Another aspect in the development of the new impactor was to provide low blank values and minimum cross-contamination between subsequent sampling periods, as explained in lines 64 and 65 (introduction) of the preprint manuscript. Several constructive measures were taken for this purpose, which are explained in lines 86 to 104 of the preprint manuscript. The effectiveness of these measures was verified by the experiments described in sections 3.1 and 3.2 and the results presented in sections 4.1 and 4.2 demonstrate very low blank values and cross-contamination for the new impactor.

Last not least, a new method for coating the sample carriers with an adhesive layer was developed and described in detail in section 2.3 of the preprint manuscript.

3.2 Why did the new impactor select PM10, PM2.5 and PM1, but not PM1, PM1-2.5 and PM2.5-10? PM0.11-1, PM1-2.5 and PM2.5-10 were showed in Figure 5, so what the impactor actually sampled?

Indeed, specific particle size fractions are separated at the individual stages of the impactor, which means that there is no single impactor stage at which the entire PM2.5 or the entire PM10 fraction is collected. Rather, the PM1, PM2.5 and PM10 fractions are calculated by summing up the mass contents of the particles impacted at the relevant individual stages.

For each stage of the impactor, particles with an aerodynamic equivalent diameter given as the separation diameter are sampled with a nominal collection efficiency of 50 %, larger particles are sampled with higher collection efficiency, and smaller particles are sampled with lower collection efficiency. For example, on stage 1, which has a separation diameter of 10 µm (Table 1), particles that are equal to or larger than 10 µm are sampled with a collection efficiency of 50 % or larger, while the majority of particles smaller than 10 µm pass through stage 1 and enter stage 2 of the impactor. At stage 2, the separation diameter is 2.5 µm (Table 1), thus collecting particles equal to or larger than 2.5 µm on the sample carrier of stage 2, and so on. Accordingly, particles larger than 2.5 µm and smaller than 10 µm are impacted on the sample carrier in stage 2, or in other words, the PM10 - PM2.5 fraction.

These proportions are visualised by the different colours of the bars in Figure 5:
the red bar represents the particle mass impacted on stage 2 (PM10 – PM2.5),
the green bar represents the particle mass impacted on stage 3 (PM2.5 – PM1),
and the blue bar represents the sum of particle masses impacted on stages 4 and 5 (adopted as PM1 with additional notes in lines 359 to 363 of the preprint manuscript).
In view of referee's comment, the authors amended section 4.2, line 366 to 369 of the

revised manuscript for clarification as follows:

> "…periods in three size fractions, i.e. PM$_1$ (top of blue bar, i.e. sum of masses of impactor stages 4 and 5), PM$_{2.5}$ (top of green bar, i.e. sum of masses of impactor stages 3, 4, and 5; the green bar represents the particle mass impacted on stage 3 corresponding to PM$_{2.5}$ – PM$_1$) and PM$_{10}$ (top of red bar, i.e. sum of masses of impactor stages 2, 3, 4 and 5; the red bar represents the particle mass impacted on stage 2 corresponding to PM$_{10}$ – PM$_{2.5}$). In addition, the…"

3.3   What elements can be detected by the TXRF? Why the Fe, Cr, and Ni mean blank values were showed in section 4.1, but the concentrations of Zn, Cu, Mn, Pb and Ni were showed in section 4.3? How about other metals, such as Co, V, As? Blank values and detection limits of all the measured elements should be summarized.

Generally speaking, elements can be detected by TXRF if fluorescence can be induced by the excitation radiation and if the induced fluorescence can "be seen" by the detector. Light elements with a low atomic number, such as carbon, are particularly difficult to measure because both efficient excitation with standard X-ray tubes and sufficient detection of the fluorescence are problematic. For particularly heavy elements such as cadmium, on the other hand, excitation with X-rays of higher energy is required. The method works optimally for elements with an order number of 14 or higher. The spectrometer used for the present study offers the possibility of using different excitation energies (see lines 260 to 263 of the preprint manuscript), with which the heavy metals of interest for the present study could be detected very well.

In section 4.1, the blank values of Fe, Cr and Ni were determined because the new impactor was manufactured from stainless steel, i.e. from a material that essentially consists of Fe, Cr and Ni, and because very high blank values of the impactor material were previously reported in the literature (Klockenkämper et al.). Numerous measures to prevent particle abrasion and adhesion were implemented in the design of the new impactor (lines 98 to 104 of the preprint manuscript), and their effectiveness was investigated in the present study. Therefore, the blank values of the elements Fe, Cr and Ni of the impactor material stainless steel were analysed and shown in section 4.1.

In Section 4.3, on the other hand, the results of particles collected in outdoor air were presented using the elements Zn, Cu, Mn, Pb and Ni as examples in order to demonstrate the possibilities and current limits of the analysis options with the new impactor under real conditions in the field. The corresponding blank values measured as part of the analyses for section 3.1 are 0.001 ng for Zn and Cu, 0.003 ng for Pb, and 0.002 ng for Ni, thus well below the values measured in outdoor air.

For the atmospheric conditions at the time (29 August 2022) at the sampling location (Potsdamer Platz, Berlin), the concentration was sufficiently high for some elements (Zn, Cu, Mn) to be detected by taking a 30-minute sample at 5 slm, while other elements (e.g. As) could not be detected (see line 355 of the preprint manuscript). For elements present at very low concentrations, the sampled air volume would have to be increased, for example by using a sampling period longer than 30 minutes.

3.4   The concentrations can largely changed in the environment. How to set the sampling time to ensure that the sample meets the needs of the analysis and is not overloaded?

This comment by the referee applies to impactor sampling in general. In practice, expected concentrations of the chemical elements of interest can often be estimated from previous studies. This information can be used to calculate the expected optimum sampling time.

3.5   Measurement results using this impactor and those using commercial impactors should be compared, to show the improvement of this new impactor.

A direct comparison with other commercially available impactors has not been performed. A quantitative comparison of chemical composition using TXRF would be difficult because the deposition patterns of these impactors have lateral dimensions in some stages that lie outside the excitation range and/or the detection range of the TXRF spectrometer used in the present study. However, the mass concentrations determined with the new impactor and shown in section 4.3 are consistent with the annual mean concentrations of Pb in PM10 (4.4 ng/m$^3$ in 2021) and Ni in PM10 (0.6 ng/m$^3$ in 2021) measured at the Berlin air quality network BLUME station Berlin-Neukölln.

Regarding improvement of the impactor, the authors are not aware of a commercial impactor that allows the determination of the concentration of metals in atmospheric aerosol particles using a battery-operated, 30-minute sampling process that does not require an external energy supply and is portable and mobile.

---

## Author Response (AR2)

**Response to Report #2 of referee #3**

We would like to thank Referee #3 for his continued support and comments.

We respond to the specific comments in the following:

*1) Although the authors tried to response the comments, the manuscript were poorly revised. For example, it was suggested to summarize the problems of previous instruments and then state the advances of this study in the revised introduction and conclusion, rather than only response.*

In view of this comment of referee #3, the authors further revised the manuscript (1) to summarize the problems of presently available impactors when using TXRF analysis of aerosol metal content and (2) to emphasize the advances of the newly developed cascade impactor in the correspondingly revised introduction;

see section 1, line 56 and 57 of the downloaded preprint manuscript:
> "…the classifying nozzles deposit the particles is usually significantly larger than the area analysed by TXRF. *As a result, only a fraction of the particles collected by the impactor are actually analysed, while a significant proportion of the impacted particles remain inaccessible for the TXRF analysis and consequently, the overall sensitivity is reduced. To take…"*

and section 1, line 65 of the downloaded preprint manuscript:
> "…*provide low blank values and minimum cross-contamination between subsequent sampling periods. The impactor nozzles of the new cascade impactor are arranged in such a way that all particles impacted on the respective sample carrier contribute to the TXRF analysis, thus increasing the overall sensitivity. This enables shorter sampling times, which in turn opens up new possibilities for identifying pollutant sources. The corresponding arrangement of the impactor nozzles could not be achieved simply by compressing a previous deposition pattern, but rather the number and lateral arrangement of the impactor nozzles have to be recalculated as well as the diameters of the impactor nozzles. As a result of these considerations, the new cascade impactor is designed for a reduced gas mass flow compared to commercially available impactors, which in turn enables the use of smaller pumps, and thus portable and mobile battery-powered operation of the impactor in the field. We will first present the…"*

In our opinion, the Conclusions section (section 5) already reflects all relevant aspects of the progress achieved by the new impactor presented in the manuscript.

*2) The authors responded that "A quantitative comparison of chemical composition using TXRF would be difficult because the deposition patterns of these impactors have lateral dimensions in some stages that lie outside the excitation range and/or the detection range of the TXRF spectrometer used in the present study." How can the authors ensure the accuracy of this new instrument? If the quantitative comparison can not be conducted, how can the instrument be applied worldwide? I think parallel filters can be sampled and analyzed to conduct the quantitative comparison.*

Quantification of impactor samples by TXRF is inherently difficult and still a topic of active research. For example, Seeger et al. (2021) show the potential and limitations of the quantification of element mass concentrations in ambient aerosol samples using a commercial cascade impactor and TXRF in a comprehensive study. Vigna et al. (2022) investigate the influence of impactor deposition patterns on TXRF analysis in a related study. Further examples of current studies on quantitative (T)XRF analysis are included in the manuscript, e.g. Hönicke et al. (2019).

Our manuscript focuses on the development of a new impactor design in combination with established TXRF analysis and first applications. We show that the quantitative results for lead and nickel in PM10 are consistent with annual mean values of PM10 concentrations of lead and nickel in the state monitoring network. We acknowledge that this is not a direct comparison of impactor and filter samples as suggested by the referee. A direct comparison of the new impactor design with commercial impactors would be difficult because of the influence of different impactor deposition patterns on the TXRF analysis. Filter samples, as suggestes, cannot be directly analyzed with the TXRF used in our study. A direct quantitative comparison of our impactor samples and filter samples would require extraction and subsequent analysis, for example by ICP-MS, which is beyond the scope of our study.

Hönicke, P., Andrle, A., Kayser, Y., Nikoaev, K.V., Probst, J., Scholze, F., Soltwisch, V., Weimann, T., & Beckhoff, B. (2020). Grazing incidence-x-ray fluorescence for a dimensional and compositional characterization of well-ordered 2D and 3D nanostructures. Nanotechnology, 31(50), 505709, https://doi.org/10.1088/1361-6528/abb557

Seeger, S., Osan, J., Czömpöly, O., Gross, A., Stosnach, H., Stabile, L., ... & Beckhoff, B. (2021). Quantification of element mass concentrations in ambient aerosols by combination of cascade impactor sampling and mobile total reflection X-ray fluorescence spectroscopy. *Atmosphere*, *12*(3), 309.

Vigna, L., Gottschalk, M., Cacocciola, N., Verna, A., Marasso, S.L., Seeger, S., Pirri, C.F., Cocuzza, M. (2022) Flexible and reusable parylene C mask technology for applications in cascade impactor air quality monitoring systems. Micro and Nano Engineering 14, 100108.